# Intranasal Administration of Forskolin and Noopept Reverses Parkinsonian Pathology in PINK1 Knockout Rats

**DOI:** 10.3390/ijms24010690

**Published:** 2022-12-30

**Authors:** Ruben K. Dagda, Raul Y. Dagda, Emmanuel Vazquez-Mayorga, Bridget Martinez, Aine Gallahue

**Affiliations:** 1Department of Pharmacology, Reno School of Medicine, University of Nevada, Reno, NV 89557, USA; 2CNS Curative Technologies LLC, 450 Sinclair Street, Reno, NV 89501, USA

**Keywords:** Parkinson’s disease, intranasal administration, BDNF, PKA, Forskolin, therapeutics, disease-modifying

## Abstract

Parkinson’s Disease (PD) is a brain-degenerative disorder characterized by a progressive loss of midbrain dopamine neurons. Current standard-of-care includes oral administration of Levodopa to address motor symptoms, but this treatment is not disease-modifying. A reduction in Protein Kinase A (PKA) signaling and neurotrophic support contributes to PD pathology. We previously showed that enhancing PKA activity in the brain via intraperitoneal administration of Forskolin in Parkinsonian rats (PINK1 knockout) abrogate motor symptoms and loss of midbrain dopamine neurons. Given that intraperitoneal administration is invasive, we hypothesized that intranasal administration of Forskolin and a second nootropic agent (Noopept) could reverse PD pathology efficiently. Results show that intranasal administration of a formulation (CNS/CT-001) containing Forskolin (10 µM) and Noopept (20 nM) significantly reversed motor symptoms, loss of hind limb strength, and neurodegeneration of midbrain dopamine neurons in PINK1-KO rats and is indistinguishable from wild-type (WT) rats; therapeutic effects associated with increased PKA activity and levels of BDNF and NGF in the brain. Intranasal administration of CNS/CT-001, but not Forskolin, significantly decreased the number of α-synuclein aggregates in the cortex of PINK1-KO rats, and is indistinguishable from WT rats. Overall, we show proof of concept that intranasal administration of CNS/CT-001 is a non-invasive, disease-modifying formulation for PD.

## 1. Introduction

Parkinson’s Disease (PD) is a chronic brain degenerative disorder that affects approximately 1% of the worldwide population [1,2], with a prevalence expected to double significantly by 2040 [3]. The economic impact of PD is significant, with an estimated cost of treatment for patients projected to exceed more than $14.4 billion a year and up to $79 billion by 2037 [4,5]. Worldwide, a notable increase in the incidence of PD has surged significantly in the United States and Norway compared to other countries for the past 30 years [6]. These abysmal statistics underscore the need to develop new therapies to alleviate symptoms and stop disease progression. Motor symptoms are exhibited in PD patients when the majority of *substantia nigra* (SN) neurons degenerate (>90%) [7]. Motor symptoms of PD include resting tremors, the rigidity of the limbs, postural instability, and decreased gait and balance, which typically develop in one limb of the patient, and eventually manifest in all limbs [1,7]. Non-motor symptoms include chronic fatigue, constipation, and progressive cognitive decline in patients with advanced PD [8,9,10]. In addition, a subpopulation of PD patients can manifest psychiatric disorders and conditions, including anxiety, altered sleep patterns, and major clinical depression [11,12,13]. Approximately 10% of all PD cases involve gene mutations. However, greater than 85% of PD cases have no known etiology (idiopathic) [14,15,16]. Mutations in more than 20 genes that encode for proteins that modulate mitochondrial structure/function, mitochondrial turnover (mitophagy), lysosomal function, ubiquitin-proteasome pathways, cytoskeletal dynamics, and neurotrophic support have been associated with juvenile-onset recessive or autosomal dominant forms of Parkinsonism [16,17,18,19,20].

Standard-of-care for PD consists of oral or intramuscular administration of Levodopa to abrogate motor symptoms by elevating the endogenous level of dopamine in the brain. However, current PD treatment only addresses clinical symptoms of PD without any reversal of degeneration of midbrain dopamine neurons and the cortex [21,22]. For those patients that are not responsive to Levodopa administration, other standard-of-care therapies include dopamine receptor agonists, anticholinergics agents, inhibitors of catechol o-methyl transferase (COMT), apomorphine, and inhibitors of monoamine oxidase B (MAO-B) [23,24,25]. However, these treatments can induce significant unwanted effects, including vomiting, nausea, reduced blood pressure, hallucinations, and uncontrolled motor movements (dyskinesias) [21,26]. New anti-PD therapeutics developed in the pipeline and tested clinically include the administration of neurotrophic factors, mitochondrial-directed antioxidants, gene therapies, and the repurposing of existing FDA-approved drugs [27]. However, many anti-PD therapies under development can carry inherent risks and have not been shown to have significant clinical benefits compared to standard-of-care (e.g., Levodopa). Therefore, it is highly urgent to develop non-invasive, disease-modifying agents that can achieve high patient compliance.

Mitochondrial dysfunction plays a strong role in contributing to PD pathology. PTEN-induced kinase 1 (PINK1), a large serine/threonine (ser/thr) kinase targeted to the mitochondrion and the cytosol, is mutated in recessive familial forms of PD. PINK1 regulates mitochondrial structure/function, mitochondrial turnover (mitophagy), and synaptic plasticity of neurons via downstream activation of Protein Kinase A (PKA) [23,28,29,30,31,32,33,34]. The fact that mutations in PINK1 are associated with familial forms of PD suggests that disruption in kinase signaling in the mitochondrion contributes to the neurodegeneration of midbrain dopamine neurons in the SN of PD patients [35,36]. Additionally, mitochondrial-derived oxidative stress caused by dysfunctional mitochondria can lead to impairment of the ubiquitin-proteasome system and the subsequent misfolding, and overt accumulation of α-synuclein, which in turn contributes to the formation of large intracellular, proteinaceous aggregates known as Lewy bodies [18,37]. An overt intracellular accumulation of Lewy bodies in the midbrain, and subsequent spread to the cortex contribute to cognitive decline in PD patients due to increased oxidative stress, decreased levels of antioxidants, mitochondrial dysfunction, and failure in neuronal lysosomal-autophagosomal systems [37,38,39,40]. In addition to mitochondrial dysfunction, a reduction in pro-survival PKA signaling and neurotrophic support mediated by Brain-Derived Neurotrophic Factor (BDNF) and Nerve Growth Factor (NGF) in the brain contribute to PD pathology [41,42,43]. In cultured primary cortical and midbrain dopamine neurons and PINK1 knockout murine models, our research group discovered that PINK1-deficient neurons showed a substantial decrease in the global activity of PKA, a well-documented ser/thr kinase that governs important neuronal functions, including neuronal development, plasticity, memory consolidation, and survival. Pathologically, this decrease in PKA activity contributes to mitochondrial dysfunction, mitochondrial fragmentation (fission), increased oxidative stress, loss of dendritic arbors, depletion of dendritic mitochondria, the onset of motor symptoms of PD, and neurodegeneration of midbrain dopamine neurons [41,44,45,46].

Forskolin is a small cyclic diterpenoid isolated from the Southwest Asian plant *Coleus forskohlii*, [47,48]. Importantly, different extracts with a low concentration of Forskolin (20–30%) are safe, oral weight-loss supplements commercially available worldwide that induce modest benefits in humans, including weight loss [49]. Additionally, Forskolin has been considered as a neuroprotective agent [50] due to its ability to stimulate neurogenesis, enhance mitochondrial fusion and function, and elevate prosurvival signaling as a consequence of elevating cyclic AMP-dependent PKA activity [44,51,52]. In further support of its ability to exert neuroprotective effects, we have shown that intraperitoneal administration of Forskolin (five doses over ten days) in vivo is sufficient for ameliorating motor symptoms of PD and corporeal fatigue as assessed by measuring hind limb strength and reverses degeneration of midbrain dopamine neurons in symptomatic PINK1-KO rats [46]. In that preclinical study [46], we showed proof of principle that elevating PKA activity reduces motor symptoms and loss of midbrain dopamine neurons in PINK1-KO rats, a genetic model of PD. Like Forskolin, N-phenylacetyl-L-prolylglycine ethyl ester (Noopept) is a safe, nootropic compound that is commercially available to humans to enhance cognitive activity [53]. Noopept is able to penetrate the blood-brain barrier (BBB) when administered orally [54]. In animal models, oral administration of Noopept has been shown to enhance the intracellular levels of BDNF by binding to the Tropomyosin Receptor Kinase B (TrkB) receptor in neurons and, thereby, induce neurogenesis. In addition to BDNF, the administration of Noopept can significantly enhance the endogenous level of the cleaved, mature form of NGF in the cortex enhancing neurogenesis and elevating cortical metabolic activity in wild-type rats [55]. However, unlike Forskolin, which shows limited potential to cross the BBB via oral administration, Noopept can efficiently penetrate the BBB in an unmodified, active form when administered orally in rats [56]. To date, the BBB poses a major technical hurdle for the development of new anti-PD therapies, given that it prevents the ability of drugs to accumulate in target regions of the brain. This pharmacological hurdle complicates the management and patient compliance with brain-degenerative diseases. In order to increase patient compliance, bypass the BBB to attain high brain bioavailability, and avoid extensive metabolic inactivation of drugs in the liver and GI tract, the intranasal delivery of pharmacologically active compounds has been recently recognized as a promising and viable therapy for treating brain-degenerative disorders by bypassing the BBB [57,58].

To bypass the BBB and achieve high bioavailability, we developed a new intranasal formulation that contains analogs of Forskolin and Noopept, termed CNS/CT-001 (Central Nervous System/Curative Technologies parent formulation 001), with the goal of offering a non-invasive pharmaceutical alternative for treating motor symptoms PD, and with the potential to achieve high patient compliance while attaining high brain bioavailability. Indeed, several anti-Parkinson’s disease standard-of-care therapeutics have been reformulated for intranasal delivery to treat motor symptoms of PD, including intranasal Levodopa to address the “off” episodes in PD patients taking oral Levodopa.

Approximately 50% of PD patients experience significant cognitive decline and loss of brain energy in part due to an overt accumulation of Lewis bodies in the midbrain and motor cortex, which contribute to the progressive degeneration of midbrain dopamine neurons and the cortex. To date, there are no effective treatments that can reverse cognitive decline in PD or Lewis Body Dementia (LBD) patients. The use of Memantine, an NMDA receptor antagonist, used to treat cognitive decline in Alzheimer’s Disease (AD), only exerts mild effects in reducing cognitive decline in PD patients [59]. In a concerted effort to find treatments to help alleviate cognitive decline in brain-degenerative diseases, several nootropics have been developed and indicated for medical use in Europe and East Asia to address the age-related cognitive decline and treat mild cognitive decline induced by AD and major clinical depression. These nootropics include Modafinil, Piracetam, and Noopept, small synthetic peptides that are commercially available that have been developed for bypassing the BBB [53,60]. Experimentally, Noopept has been characterized as a nootropic agent capable of increasing cognitive function in cell culture and in vivo models of brain-degenerative diseases by clearing out protein aggregates from brain tissue. In several in vitro and in vivo models of AD, the administration of Noopept can efficiently target oligomeric β-amyloid by stimulating the production of anti- β-amyloid (25–35) antibodies, presumably stimulating the clearance of β-amyloid aggregates [61]. In vitro, NMR and cell culture studies have revealed that Noopept can efficiently reduce the level of oligomeric forms of α-synuclein and thereby improves neuronal survival against α-synuclein mediated cytotoxicity [62]. To date, the neuroprotective effects of the combined intranasal delivery of nootropic agents Forskolin or Noopept have not been explored in animal models of PD. Given that we have recently published that pharmacological-mediated enhancement of PKA signaling via intraperitoneal delivery of Forskolin is sufficient to ameliorate Parkinson’s pathology in 10-month-old PINK1-KO rats [46], in this study, we hypothesized that intranasal delivery of more than one nootropic agent that can enhance both PKA and neurotrophic support (BDNF + NGF) could yield neuroprotective effects in PINK1-KO rats in an additive manner compared to intraperitoneal administration of Forskolin alone [46]. Secondly, given its ability to clear out oligomeric forms of α-synuclein [62], we also hypothesized that including Noopept in conjunction with Forskolin in the same formulation would facilitate the clearance of α-synuclein aggregates in the brain during the treatment of Parkinsonian rats. The rationale for combining Noopept and Forskolin in the same formulation is that both nootropic agents can, directly and indirectly, enhance PKA signaling and stimulate the level of neurotrophic support (BDNF + NGF) to perpetuate neuroprotective signaling in the brain in a positive feedback manner. Unlike intraperitoneal administration, another rationale for intranasal delivery of these nootropic agents is that direct nose-to-brain targeting is a non-invasive therapeutic alternative to achieve high bioavailability of active ingredients (Forskolin and Noopept) in the brain to reverse PD pathology in PINK1-KO rats.

The overarching goal of the present study is to determine the extent by which pharmacologically elevating both neuroprotective PKA signaling and BDNF levels in the midbrain via intranasal delivery of Forskolin and Noopept can reverse PD pathology in symptomatic Parkinsonian rats (PINK1 knockout) [63,64]. Overall, our results show that intranasal administration of CNS/CT-001 can reverse motor symptoms of PD, the loss of hind limb strength, reduce the amount of α-synuclein aggregates in the cortex and significantly reverse neurodegeneration of midbrain dopamine neurons in symptomatic PINK1-KO rats. Mechanistically, the therapeutic benefits exerted via intranasal administration CNS/CT-001 are associated with significantly increased PKA activity, high brain bioavailability of the active ingredients (Forskolin and Noopept), and increased levels of the neurotrophic factors, including NGF and BDNF in the brain.

Overall, our preclinical data shows that intranasally delivery of CNS/CT-00—which contains two nootropic agents- in Parkinsonian rats is a non-invasive disease-modifying therapy for treating PD for eliciting neuroprotective signaling and neurotrophic support in the brain.

## 2. Results

The PINK1-KO rat is an excellent model of PD for performing clinical translational research as it recapitulates many clinical symptoms of PD in humans, including motor symptoms and decreased hind limb muscle strength, a proxy of non-motor symptoms [45,46,64]. By employing biochemical assays in this in vivo PD model, one can measure clinical indicators of PD progression, including significant reductions in brain energy (oxidative phosphorylation + glycolysis) in the brain, a loss of dopamine neurons in the SN, as well as analyzing the accumulation of *α*-synuclein in the cortex, a proxy for Lewy body aggregates [45,46,64]. PINK1-KO rats also show significant reductions in the level of cytosolic and mitochondrially-localized antioxidants in the midbrain (superoxide dismutase 2 and catalase), decreased PKA activity in the midbrain, and a significant reduction in the intracellular levels of BDNF [45,64,65]. Given that we have recently published that pharmacological-mediated enhancement of PKA signaling via intraperitoneal delivery of Forskolin is sufficient to ameliorate Parkinson’s pathology in 10-month-old PINK1-KO rats [46], we surmised that intranasal delivery of CNS/CT-001 (a formulation that contains Forskolin and Noopept) could exert more significant neuroprotective effects in PINK1-KO rats by pharmacologically elevating the activity of PKA and intracellular levels of BDNF in the brain and reduce the level of α-synuclein aggregates. In addition, we surmised that intranasal administration of CNS/CT-001 could significantly increase brain bioavailability of the active ingredients (Forskolin and Noopept) with favorable pharmacokinetics. To test these hypotheses, we analyzed the therapeutic effects of intranasal administration of CNS/CT-001 on motor symptoms, hind limb strength, accumulation of α-synuclein aggregates in the brain, and loss of SN neurons in PINK1-KO rats by using a myriad of well-validated behavioral and neurochemical assays as previously published by our research group [46,63].

### 2.1. Intranasal Administration of Forskolin Is Sufficient to Reverse Motor Symptoms and Hind Limb Fatigue in PINK1-KO Rats

To test our hypotheses, we assessed the effects of intranasally administering symptomatic PINK1-KO rats with vehicle control (saline solution, pH 7.4), Forskolin alone (10 µM), or with CNS/CT-001 (Forskolin 10 µM and 20 nM Noopept) on motor coordination and muscle fatigue. By using the beam balance test, we noticed a significant elevation in motor symptoms in 10-month-old PINK1-KO rats compared to untreated wild-type (WT) Long Evans hooded rats as assessed by quantifying the mean number of foot slips and falls for each animal crossing a 1-m beam balance (Figure 1). However, intranasal delivery of either Forskolin or CNS/CT-001 (5 doses for ten days) in PINK1-KO rats significantly abrogated motor symptoms of PD and displayed a similar aggregate motor score as untreated wild-type rats (Figure 1).

The data suggests that intranasal administration of Forskolin yields similar anti-Parkinsonian effects as intraperitoneal administration of Forskolin in PINK1-KO rats [46]. In addition, there were no significant differences in motor symptoms noted between PINK1-KO rats intranasally treated with either Forskolin or with CNS/CT-001. These observations suggest that intranasal administration of Forskolin is sufficient to reverse motor symptoms, whereas the administration of Noopept does not confer a significant additive effect (Figure 1).

Next, we surmised that the intranasal administration of Forskolin or CNS/CT-001 could reverse the loss of hind limb strength in PINK1-KO rats in a comparable manner to the intraperitoneal administration of Forskolin [46]. Untreated PINK1-KO rats showed a 30% reduction in the mean hind limb strength (g/kg of force) compared to wild-type rats. In contrast, we observed that intranasal delivery of either Forskolin or CNS/CT-001 completely abrogated the loss of hind limb strength in PINK1-KO rats to a similar extent as untreated wild-type rats (Figure 2). These data suggest that intranasal administration of Forskolin can significantly exert anti-Parkinsonian effects to a similar extent as intraperitoneal administration of Forskolin in 10-month-old PINK1-KO rats [46]. However, we did not observe any statistically significant differences between vehicle-treated WT rats vs. PINK1-KO rats treated with Forskolin or with CNS/CT-001, or between PINK1-KO rats intranasally-treated with Forskolin or with CNS/CT-001 suggesting that intranasal administration of Forskolin is sufficient to reverse the loss of hind limb strength in 10-month-old symptomatic PINK1-KO rats, whereas intranasal administration of Noopept does not confer an additive effect (Figure 2).

While an extensive toxicological report is warranted in the future, it is worth noting that the intranasal delivery of CNS/CT-001 did not cause any local irritation, itching, or inflammation of the nostrils or nasal passages of rats (data not shown), and did not cause any noticeable adverse events (e.g., noticeable loss of weight, discoloration, and loss of fur, or death). Consistent with these observations, we have previously shown that intraperitoneal administration of Forskolin does not cause significant weight loss in PINK1-KO rats treated for more than five weeks [46].

### 2.2. Intranasal Administration of CNS/CT-001 Reverses α-Synuclein Aggregation in the Brain

The accumulation of Lewy bodies throughout the midbrain and cortex is a pathological hallmark of PD, which contributes to cognitive decline and the onset of dementia in a subset of advanced PD patients. Given that symptomatic PINK1-KO rats are characterized by a widespread accumulation of α-synuclein aggregates in the brain—a proxy of Lewy body accumulation-, we then investigated the extent that intranasal administration of Forskolin or with CNS/CT-001 can reduce the accumulation of α-synuclein aggregates in the brain. By performing IHC, we quantified the amount of α-synuclein accumulation by counting the number of large α-synuclein puncta normalized to the number of cells in slices of the prefrontal cortex (PC) derived from PINK-KO rats that were intranasally treated with Forskolin, vehicle control or with CNS/CT-001. In brief, as reported by other research groups [66], we observed a two-fold increase in the number of α-synuclein aggregates in the PC of 10-month-old PINK1-KO rats compared to untreated wild-type rats (Figure 3A,B). In contrast, while the intranasal administration of Forskolin showed a non-significant trend in reducing the mean number of α-synuclein aggregates in the motor cortex (M3 and M4 regions), intranasal delivery of CNS/CT-001 was able to significantly reduce the level of α-synuclein aggregates in the cortex to a similar extent as untreated wild-type rats (Figure 3). Overall, this data suggests that the combined intranasal delivery of Forskolin and Noopept can induce an additive effect in significantly reducing the level of α-synuclein aggregates in symptomatic PINK1-KO rats compared to intranasal administration of Forskolin (Figure 3).

### 2.3. Intranasal Administration of Forskolin Reverses the Loss of SN Neurons in Symptomatic PINK1-KO Rats

We have previously published that intraperitoneal administration of Forskolin is sufficient to reverse the degeneration of midbrain dopamine neurons in symptomatic PINK1-KO rats [46]. Next, we surmised that intranasal delivery of Forskolin or CNS/CT-001 could reduce the loss of midbrain dopamine neurons in symptomatic PINK1-KO rats. To this end, we performed an IHC-mediated analysis of midbrain dopamine neurons in the SN of WT and PINK1-KO rats intranasally treated with vehicle alone, with Forskolin, or with CNS/CT-001. Consistent with previous studies, IHC analysis of midbrain dopamine neurons in SN slices showed that 10-month-old PINK1-KO rats displayed a significant loss of midbrain dopamine neurons in the SN, as evidenced by a decrease in TH-specific immunofluorescence in SN slices normalized to the number of cells, compared to age-matched WT rats (Figure 4A,B) [46,65,67]. However, intranasal delivery of Forskolin or with CNS/CT-001 significantly restored the TH staining in the SN of symptomatic PINK1-KO rats to a similar extent as observed in age-matched vehicle-treated wild-type rats (Figure 4A,B). It is worth noting that intranasal administration of CNS/CT-001 or Forskolin was equally effective in reversing the loss of TH fluorescence in PINK1-KO rats, suggesting pharmacological activation of PKA by intranasal delivery of Forskolin is sufficient to reverse neurodegeneration of midbrain dopamine neurons (Figure 4A,B).

### 2.4. Administration of CNS/CT-001 Enhances Neuroprotective PKA Signaling in the Brain

Next, to determine the extent to which the anti-PD effects induced by intranasal administration CNS/CT-001 (Figure 1, Figure 2, Figure 3 and Figure 4) correlate with enhanced brain PKA activity, we performed ELISA-based PKA activity assays as published [41]. We evaluated PKA activity in the cortex of WT rats intranasally treated with a single dose of Forskolin at increasing concentrations for 24 h. Consistent with its ability to increase PKA activation, we observed more than a three-fold increase in PKA activity in the cortex of rats intranasally treated with Forskolin at increasing concentrations compared to vehicle-treated 10-month-old WT rats (Figure 5). These biochemical data suggest that intranasal administration of Forskolin, contained in CNS/CT-001, can efficiently penetrate the BBB to activate PKA activity in the cortex of wild-type rats (Figure 5).

### 2.5. Forskolin and Noopept Accumulate in the Cerebral Spinal Fluid at High Levels When Administered Intranasally

We then performed a pharmacokinetic analysis of Forskolin and Noopept in WT rats intranasally treated with CNS/CT-001 to further corroborate the ability of Forskolin to reach the brain. In brief, we performed HPLC-mediated analysis of Forskolin levels in the serum and cerebrospinal fluid (CFS) at various time points in WT rats that were intranasally treated with CNS/CT-001. We observed a significant increase in Forskolin levels in the CSF within 30 min of intranasal administration, which plateaued at 4 h (Tmax), followed by a significant reduction at 24 h of treatment (Appendix A). In the serum, HPLC-mediated analysis showed a significant increase of Forskolin within 1 h of administration and a Tmax of 4 h, which coincided with a concomitant increase in the CSF from rats that were intranasally treated with CNS/CT-001 (Appendix A). This data suggests that Forskolin contained in CNS/CT-001 is sufficient to efficiently bypass the BBB and achieve sufficient bioavailability when administered intranasally (Appendix A) to elevate PKA activity (Figure 5). Indeed, further analysis showed that more than 33% of the total mass of Forskolin contained in CNS/CT-001 is distributed to the CSF when delivered intranasally. This data suggests that intranasally-administered Forskolin can penetrate the BBB with sufficient bioavailability. Given the lack of antibodies and biochemical assays to measure Noopept directly in both blood and CSF, we next measured the level of Noopept by HPLC in the serum and CSF from WT rats that were intranasally treated with CNS/CT-001. However, although we were able to measure Forskolin successfully via HPLC (Appendix A), we lacked sufficient serum and CSF samples (>200 µL) to measure Noopept within the biological samples from the same cohort of rats. Hence, to circumvent this problem, we intranasally treated New Zealand rabbits with CNS/CT-001 for 24 h and measured the level of both Forskolin and Noopept in the serum and CSF samples collected at different time points. The justification for employing New Zealand white rabbits for determining the pharmacokinetics of Noopept is their significantly larger body size compared to rats which are highly amenable to collecting a larger amount of blood and CSF that can be extracted in a humane manner. As in rats, we detected Forskolin in the CSF within 1 h in intranasally-treated rabbits with a Tmax of 10 h (Appendix A), whereas Forskolin was detected at a very low concentration in the serum at the same time point (Appendix A). Like in rats, approximately 30% of the starting material of intranasally administered Forskolin in CNS/CT-001 was detected in the CSF, suggesting that Forskolin can bypass the BBB efficiently in rabbits. In addition, Noopept was detected in the CSF of rabbits within 1 h of administration and also showed a Tmax of 10 h (Appendix A). It is worth noting that Noopept levels were undetectable in the serum of intranasally-administered rabbits (Appendix A), presumably since the majority (~80%) of the total mass of intranasally-administered Noopept contained in CNS/CT-001 reached the CSF (Appendix A).

While an extensive toxicological report is warranted in the future for CNS/CT-001, it is worth noting that descriptive observations suggest that intranasal delivery of CNS/CT-001 did not cause any local irritation or inflammation of the nostrils or nasal passages of rabbits that can be expected by prolonged sneezing or itching (data not shown), and did not cause any noticeable adverse events (e.g., loss of weight, discoloration of fur, or death).

Overall, our compiled pharmacokinetic data in both rats and rabbits suggest that Forskolin and Noopept can bypass the BBB efficiently and achieve high bioavailability in the brain when administered intranasally.

### 2.6. Intranasal Administration of CNS/CT-001 Elevates Levels of Neurotrophic Factors in the Brain

Given that one of the physiological effects of Noopept is enhancing the level of neurotrophic factors (NGF and BDNF) in the brain [55], we next assessed the extent to which Noopept, contained in CNS/CT-001, can affect the endogenous levels of NGF and BDNF in the brains of intranasally-treated 10-month-old PINK1-KO rats. By performing a Western blot analysis of whole brain lysates (cortex), we observed a non-significant decrease in the level of cleaved, mature form of NGF in PINK1-KO rats compared to WT rats (Appendix A). On the other hand, intranasally treating PINK1-KO rats with CNS/CT-001 induced a significant elevation in the level of mature forms of both NGF and BDNF in the frontal cortex compared to PINK-KO rats treated with vehicle (Appendix A), and a non-significant increase compared to PINK1-KO rats treated with Forskolin (Appendix A). Similar results were observed in PINK1-KO rats treated with CNS/CT-001 via intraperitoneal administration in terms of significantly elevating cleaved to uncleaved BDNF and NGF levels in the brain (data not shown). Mechanistically, our Western blot data show that intranasal administration of CNS/CT-001 can exert neurotrophic support in the brain by significantly increasing the intracellular level of BDNF and NGF, presumably to mediate anti-Parkinsonian effects, including reversing motor symptoms and the loss of midbrain dopamine neurons.

## 3. Discussion

PD is a prevalent age-related brain-degenerative disease that affects approximately 1% of all humans that are 50 years of age or older, and 2.5% of humans beyond 70 years of age [68,69]. Currently, there is no treatment that can reverse the progression of PD. The oral or intranasal administration of Levodopa is the current standard of care that temporarily alleviates motor symptoms of PD in humans [70]. However, this treatment is not disease-modifying, as PD patients often become resistant and unresponsive to the therapeutic effects of Levodopa over time. In addition, oral or intranasally-administered Levodopa leads to a myriad of side effects, including dyskinesias, nausea, loss of appetite, and hallucinations which can severely decrease the quality of life in PD patients [70]. Additionally, due to its rapid distribution in the serum and brief therapeutic duration (~2–3 h), PD patients must take multiple doses of oral Levodopa each day to maintain therapeutic efficacy, depending on disease progression/state. This rigorous and inconvenient therapeutic regimen often results in low patient compliance and inconsistent symptom management leading to unwanted on/off phases and inconsistent therapeutic efficacy. Therefore, the side effects induced by the current standard of care underscore the need for developing disease-modifying treatments that are safe, tolerable, and long-lasting in order to increase patient satisfaction and quality of life.

There are several promising anti-PD therapies in development to help ameliorate motor symptoms and overcome the loss of dopamine, including stem cell therapies, adenoviral administration of neurotrophic agents, intranasal administration of insulin, mitochondrial-targeted antioxidants (e.g., MitoQ) and intranasal formulations of Metformin. However, many anti-PD therapeutics under development are not disease-modifying and have failed to reach Phase 3 clinical trials [71,72,73].

A decrease in PKA activity and neurotrophic signaling in the SN is causally linked to a reduction in dopamine, mitochondrial dysfunction, and neurodegeneration of midbrain dopamine neurons as assessed in chemical models of PD and postmortem brain tissue of PD patients [41,46,74,75]. For instance, a decline in neuroprotective PKA activity and neurotrophic support (e.g., BDNF) in the midbrain was observed in both PINK1-KO mice and rats, contributing to PD pathology [41,45,46]. In addition, a significant decrease in the level of the cleaved mature form of BDNF in the midbrain and prefrontal cortex has been reported in 3.5-month-old PINK1-KO rats and in the SN from postmortem midbrain tissue of PD patients [42,45,46]. While the use of either exogenous BDNF or GDNF has been proposed as a therapeutic alternative for treating PD by restoring neurotrophic support in the brain [76,77,78], it is important to note that these neurotrophic factors have a low diffusion rate and therefore are limited in their ability to cross the BBB [79]. Indeed, exogenous administration of neurotrophic factors is very invasive, requires intracranial surgery, and is expensive. The intracranial infusion of GDNF, while successful in phase I clinical trials, failed to significantly improve motor symptoms in most phase II clinical trials, possibly due to the low ability of neurotrophic factors to cross the BBB and reach the SN in significant amounts to exert a therapeutic effect [76,80]. Additionally, it is worth noting that attempting to rescue degenerating midbrain dopamine neurons via intracranial administration of neurotrophic factors may not exert a therapeutic benefit given that BDNF receptors (TrkB) at the cell membrane are eventually lost in degenerating neurons as the disease progresses in PD patients, thereby impeding its mechanism of action. Therefore, it is expected that any neuroprotective responses elicited from the exogenous administration of neurotrophic factors have a limited period of effectiveness and may be restricted to the treatment of early PD [81]. Therefore, the lack of success of recombinant neurotrophic factors and other potential anti-PD therapies in the pipeline underscores an urgent need to develop new disease-modifying therapeutics that are non-invasive and have favorable brain bioavailability. In this study, our preclinical data shows that intranasal administration of a formulation that contains analogs of Forskolin and Noopept termed CNS/CT-001, can reverse PD pathology in a Parkinsonian rat model via activation of PKA, which amplifies a neuroprotective signaling pathway that is presumably downstream of cell membrane receptors (TrkB and NGF receptor), and via increasing the level of neurotrophic factors in the brain (Figure 1, Figure 2, Figure 3 and Figure 4, Appendix A). Indeed, pharmacologically elevating PKA activity in vivo via intranasal administration of Forskolin is sufficient to reverse motor symptoms and neurodegeneration of SN neurons (Figure 1, Figure 2, and Figure 5). Overall, our data is consistent with in vitro research findings from our research group, which showed that pharmacologically elevating downstream PKA activity can compensate for mitochondrial dysfunction and reverse degeneration in PINK1-deficient neuroblastoma SH-SY5Y cells and primary cortical neurons [41,44].

We have previously shown that intraperitoneal administration of Forskolin is able to efficiently reduce motor symptoms, reverse corporeal fatigue and reverse the loss of SN neurons in symptomatic 10-month-old PINK1-KO rats with long-lasting therapeutic effects [46]. Indeed, “wash out” studies showed that five doses of Forskolin over a period of 10 days significantly reduced motor symptoms for three weeks following the last dose and were able to restore hind limb strength for up to five weeks in symptomatic PINK1-KO rats [46]. In contrast, intraperitoneal administration of Levodopa was able to reduce motor symptoms for less than one week following the last dose in PINK1-KO rats, while it did not have a significant therapeutic benefit in ameliorating the loss of hindlimb muscle strength, a sign of corporeal fatigue that is characteristic of PD patients [46]. Hence, the short-term therapeutic benefits of Levodopa in symptomatic PINK1-KO rats are consistent with the on/off phases of Levodopa observed in PD patients [26]. Like intraperitoneal administration of Forskolin, our compiled behavioral and biochemical data suggest that intranasally administered Forskolin, which directly targets the brain in a non-invasive manner, exerts similar therapeutic effects but at much lower doses compared to intraperitoneal delivery of Forskolin [46] (Figure 1, Figure 2, Figure 3 and Figure 4).

It is widely recognized that one of the caveats of oral administration of Levodopa is low patient compliance, given the requirement for multiple doses per day and low bioavailability in PD patients [26]. Intranasal delivery of therapeutics has been recently recognized as a promising and viable therapeutic strategy for treating neurodegenerative disorders due to the ability to bypass the BBB [57,58]. Intranasal administration of pharmacological agents enters the BBB via three mechanisms: (1) by entering the paracellular space of the intranasal epithelium to accumulate within the cerebral spinal fluid (CSF), (2) via retrograde transport via the axons of olfactory neurons and sensory cells, or (3) by entering the bloodstream through the rich capillary beds in the upper nasal space. While intraperitoneal injection of Forskolin is able to reverse PD pathology in PINK1-KO rats [46,82], CNS/CT-001 is a formulation that was developed to enhance both neuroprotective PKA and neurotrophic support in the brain via intranasal delivery with the goal of offering a safe, non-invasive therapeutic alternative that can achieve high compliance in PD patients. CNS/CT-001 contains pharmaceutical-grade analogs of Forskolin and Noopept that are dissolved together in PBS (pH 7.4) (for details on the composition of CNS/CT-001, consult the Section 4). Noopept is a synthetic di-peptide that can enhance the levels of neurotrophic factors (BDNF and NGF) in the brain. Noopept is currently used in Europe to treat mild cognitive decline in several neurodegenerative diseases [53,55]. In this study, we show that intranasal delivery of these two active ingredients (Forskolin + Noopept) in CNS/CT-001 in symptomatic PINK1-KO rats (5 doses for ten days) significantly reduces motor symptoms (Figure 1), reverses chronic fatigue as evidenced by an increase in hind limb muscle strength (Figure 2), decreases the level of α-synuclein aggregates in the PC (Figure 3), and reverses the loss of midbrain dopamine neurons in PINK1-KO rats (Figure 4), neuroprotective effects that are associated with enhanced PKA activity (Figure 5) and elevated levels of neurotrophic factors (BDNF and NGF) in the brain (Appendix A). Importantly, intranasal delivery of CNS/CT-001 showed favorable pharmacokinetics as a single dose yielded high bioavailability when administered intranasally (Appendix A), suggesting that both compounds can successfully penetrate the BBB. In contrast, when co-administered with Carbidopa, 1–5% of orally administered Levodopa can reach the brain in humans [83,84]. Either intraperitoneal or intranasal administration of Forskolin alone is effective in reducing motor symptoms and chronic fatigue and reversing neurodegeneration in symptomatic PINK1-KO rats, suggesting that pharmacologically enhancing cyclic AMP-dependent signaling in the midbrain is sufficient to ameliorate PD pathology (Figure 1) [46]. However, our IHC data suggests that co-administering Forskolin and Noopept confers an additive effect, compared to Forskolin alone, in reducing the amount of α-synuclein aggregates in the PFC (Figure 3) and in inducing a non-significant increase in the level of BDNF compared to Forskolin treatment alone (Appendix A). Consistent with our in vivo data, the oral administration of Noopept has been shown to significantly increase the level of cleaved, mature forms of BDNF and NGF in the brain of wild-type rats [55], which have been used in aged humans to reduce age-related cognitive decline [53].

### Proposed Mechanism of Action of CNS/CT-001 in Reducing PD Pathology

Symptomatic PINK1-KO rats develop a widespread accumulation of Thioflavin S positive, proteinase resistant α-synuclein aggregates in the cortex and ventral midbrain at 12 months of age [66]. While PINK1-KO rats do not develop Lewy bodies per se, the accumulation of α-synuclein aggregates is a proxy for Lewy bodies in humans with PD and Lewy Body Dementia [85]. Indeed, the accumulation of oligomeric forms of α-synuclein in the brain is associated with cognitive decline and the onset of dementia in PD patients [85]. Our preclinical data shows that intranasal delivery of CNS/CT-001 exerts a significant reduction of α-synuclein aggregates in the PC of symptomatic 10-month-old PINK1-KO rats (Figure 4). However, the precise molecular mechanism by which intranasal administered CNS/CT-001 can mediate the clearance of α-synuclein remains to be elucidated. In symptomatic PINK1-KO rats, it is plausible that intranasal administration of CNS/CT-001 can increase the activity of the ubiquitin-proteasome or enhance macroautophagy and lysosomal activity to remove large oligomeric α-synuclein aggregates in a PINK1-independent manner. However, this is an unlikely mechanism of action since it is known that an increase in PKA activity in neurons can inhibit macroautophagy initiation, given that PKA can phosphorylate the autophagosome cargo receptor microtubule-associated protein light chain 3 (LC3) on its N-terminal domain as a mechanism to block autophagy [86,87]. In addition, an increase in BDNF levels is known to suppress autophagy and inhibit the ubiquitin-proteasome pathway. To this end, these observations rule out an increase in BDNF or PKA activation as a molecular mechanism for clearing α-synuclein aggregates via autophagy or the ubiquitin-proteasome pathway [88,89]. Given that Noopept has been characterized to reduce the number of oligomeric forms of α-synuclein in vitro [62], the most likely mechanism for reducing α-synuclein aggregates in the cortex of symptomatic PINK1-KO rats is via direct binding of Noopept with oligomers of α-synuclein via hydrophobic interactions which stimulate its conversion into a fibrillar and less pathogenic form.

Beyond stimulating neurogenesis and neuronal survival, an enhancement of intracellular PKA activity by Forskolin can exert therapeutic benefits by elevating the bioenergetic status of neurons (oxidative phosphorylation and glycolysis) in vivo [46]. In addition, enhancing PKA activity in neurons can protect chemical models of PD, presumably by reducing the level of oxidative stress derived from dysfunctional mitochondria. In vitro, treating primary hippocampal neurons or midbrain dopamine neurons with Forskolin can decrease apoptosis induced by treatment with rotenone, a complex I inhibitor used to chemically model PD [51] (US patent: US 2021/0220322).

Other possible benefits of enhancing cAMP-mediated signaling pathways in neurons through intranasal delivery of Forskolin or CNS/CT-001 may involve increasing the recruitment of NMDA receptors and GluN2A receptors in neuronal synaptic terminals. Hence, given that physiological activation of NMDA and AMPA receptors are essential for inducing long-term potentiation (LTP), memory consolidation via CREB-mediated transcription of pro-survival genes in neurons, and synaptogenesis, it is likely that an increase in cyclic AMP-dependent activity in the brain via intranasal delivery of Forskolin (or with CNS/CT-001) can lead to a maintenance of synaptic plasticity and reduce cognitive decline in PD. In addition, Noopept is known to bind directly to AMPA and NMDA receptors. This proposed mechanism of action of Noopept was previously documented in a preclinical study by Vorobyov et al., 2011. Briefly, in that study, the authors showed that oral administration of Noopept in vivo binds and selectively activates both NMDA and AMPA receptors to activate downstream neuroprotective signaling pathways, as shown in electroencephalography studies performed in rats [90]. Hence, based on this premise, it is possible that intranasal administration of CNS/CT-001 exerts additional neuroprotective effects in symptomatic PINK1-KO rats via PKA-mediated stimulation of NMDA and AMPA receptors to promote synaptogenesis and neuronal survival. However, future studies are warranted to explore this potential molecular mechanism of neuroprotection induced by CNS/CT-001.

Interestingly, the use of NMDA and AMPA antagonists is part of standard-of-care for PD. Indeed, several NMDA receptor antagonists have been used to treat non-motor symptoms of PD in patients [59]. For instance, by preventing glutamate excitotoxicity caused by NMDA receptors localized in extrasynaptic sites, Amantadine and Memantine are NMDA antagonists that have been observed to modestly reduce cognitive decline in PD patients [59]. Additionally, both NMDA and AMPA receptor antagonists have been used to treat Levodopa-induced dyskinesia in PD patients in several clinical trials [91]. On the other hand, Perampanel, an AMPA receptor inhibitor, has not been demonstrated to be clinically effective for treating Levodopa-induced motor dyskinesia in PD patients [92]. In addition, the long-term use of NMDA and AMPA receptor antagonists can produce serious side effects due to prolonged pharmacological inhibition of these receptors at synaptic sites, which are known to exert neuroprotective signaling via BDNF and CREB-mediated transcription of prosurvival gene programs [59]. Therefore, prolonged activation of synaptic NMDA and AMPA receptors will be therapeutically beneficial for PD patients as it is expected to enhance prosurvival signaling. As mentioned above, given that Noopept is known to be an agonist of both NMDA and AMPA receptors [90], it is conceivable that the long-term administration of intranasal administration of CNS/CT-001 in PD patients can ultimately be therapeutically beneficial by enhancing prosurvival signaling pathways mediated by BDNF and CREB when AMPA and NMDA receptors are activated at synaptic sites.

In addition to reducing neuroprotective PKA signaling and level of neurotrophic factors (BDNF and NGF), the progression of PD pathology involves a reduction of neurogenesis, as evident by a reduction in neural stem cells in the midbrain, which hampers the ability of the midbrain to replenish the lost dopamine neurons. Indeed, different in vivo studies have shown that neurogenesis in the midbrain and the subventricular zone of the hippocampus decrease with PD progression and further contribute to the loss of midbrain dopamine neurons in the SN and other brain areas including the prefrontal cortex [93,94,95]. Based on our in vivo data, it is conceivable that intranasal-administered Forskolin and Noopept, contained in CNS/CT-001, can ameliorate the loss of SN neurons (Figure 4) by elevating PKA activity and the intracellular levels of BDNF and NGF (Figure 5; Appendix A), neurotrophic factors known to enhance neurogenesis and stimulate an increase in neural stem cells. Therefore, it is plausible that intranasal delivery of CNS/CT-001 can enhance neurogenesis as a neuroprotective mechanism by which CNS/CT-001 reverses the loss of midbrain dopamine neurons.

Overall, this study offers preclinical evidence that CNS/CT-001 is a promising non-invasive, disease-modifying formulation that can reverse motor symptoms and disease progression, as shown by its ability to significantly reverse the loss of SN neurons in symptomatic PINK1-KO rats (Figure 4). While we recognize the intranasal delivery of Forskolin alone exerts several significant therapeutic benefits compared to CNS/CT-001, the combination of Forskolin and Noopept contained CNS/CT-001 nevertheless induces a significant beneficial effect in clearing out the α-synuclein aggregates as well as elevating the levels of neurotrophic support (BDNF + NGF), which will be critical for the maintenance of midbrain dopamine neurons in patients chronically treated with these compounds. It is possible that the production of neurotrophic factors can provide a positive feedback loop for enhancing neuronal survival and neurogenesis during long-term treatment. However, additional preclinical studies are warranted to determine the long-term benefit of intranasal administration of CNS/CT-001 (e.g., other non-motor symptoms addressed) in in vivo Parkinsonian models. Importantly, given that most cases of PD are not genetic in nature, future studies are needed to determine whether intranasal delivery of CNS/CT-001 can reverse neurodegeneration and motor symptoms in in vivo chemical models of PD (rotenone or with 6-hydroxydopamine).

## 4. Materials and Methods

### 4.1. Animal Procedures

The behavioral studies in rodents were performed in accordance with guidelines set forth by the National Institutes of Health Office of Laboratory Animal Welfare Policy, and thanks to the approval of an animal protocol (protocol #20-09-1086) via the Institutional Animal Care and Use Committee (IACUC) at the University of Nevada, Reno, and per ARRIVE guidelines. Briefly, 10-month-old WT (Charles River Laboratories International, Inc.) and PINK1-KO (Horizon Discovery) Long-Evans rats (1:1 ratio of male to female) old were employed for analyzing motor function and for performing immunohistochemistry (IHC) and neurochemical studies as further described below.

All animals were maintained 12 h light/dark cycle, with food and water ad libitum under a controlled temperature of 25–26 °C. The animals were individually caged for three days in an enriched environment before the behavioral assays per ARRIVE guidelines.

Rabbits: While we were able to perform pharmacokinetic analysis of Forskolin in rats, carrying out pharmacokinetic analysis of both Forskolin and Noopept in rats was not possible in the same cohort of animals given their small body size, and insufficient biological samples needed to analyze both Forskolin a Noopept by HPLC. To this end, we used rabbits to analyze the pharmacokinetics of intranasal-delivered Noopept and its bioavailability in the brain, given that rabbits are more amenable to collecting a larger amount of blood and CSF in a humane manner as permitted by the IACUC protocol for this study. 10-week-old New Zealand white rabbits (2.1–2.5 kg; Western Oregon Rabbit Company, Philomath, Oregon) were housed at the University of Nevada (UNR), Animal Resources Facility (Nellor Facility) as approved through IACUC protocol #20-06-1029. In addition, the state-of-the-art vivarium at UNR permits the housing and maintenance of rabbits in highly enriched environments throughout the research project. Each rabbit was housed in individual cages. Prior to the treatment of animals with pharmacological compounds (Forskolin, Noopept) intranasally, the rabbits were allowed to acclimate first for up to three days in a highly enriched environment prior to treatments with vehicle control or CNS/CT-001.

### 4.2. Motor Coordination and Grip Strength Test

WT and PINK1-KO rats intranasally treated with compounds were analyzed for motor coordination and grip strength, as further described below. Animals were recorded before (basal—“day 0”) and after the intranasal treatment with the compounds five times every other day (treated—“day 10”).

(a)
Grip strength assays


The grip strength assay is a behavioral test used to assess the muscle strength of the hind limbs and forelimbs of rodents [45,46,96]. The grip force on the front limbs and hind limbs of WT and PINK1-KO rats intranasally treated with vehicle, Forskolin, or CNS/CT-001 were assessed as previously published by our research groups using this well-validated behavioral assay [45,46,96]. It is worth noting that the maximal strength value for each rat was recorded and considered as the highest muscular force, as previously published in PINK1-KO rats [45,46].

(b)
The beam balance


To measure motor coordination in a vehicle or pharmacologically treated WT and PINK1-KO rats, we used the 1 m beam balance pre-validated by our research group by using a similar training regimen of rats, set up and parameters for the equipment as previously published [45,46,97].

For this study, the mean number of slips and falls was measured using the JWatcher^TM^ software (version 0.9) as published [45,46]. To increase rigor and eliminate bias, the number of foot slips and falls was quantified blindly by an independent observer who recorded the videos (treatments unknown) as well as analyzed the videos. We assigned a value of 1 point to each foot slip, while 2 points were assigned for each fall (value of 2). It is worth noting that both metrics were aggregated as one value known as the Total Score (TS) value, as previously published [45,46].

### 4.3. Pharmacological Treatments in Animals

Rats were subjected to intranasal administration of either vehicle alone (1:200 DMSO, at pH 7.5), with Forskolin (10 µM), or with CNS/CT-001 (10 µM Forskolin, 20 nM Noopept in PBS at pH 7.4) by administering 75 µL of each solution to each nostril two times daily.

Preparing CNS/CT-001: To prepare CNS/CT-001, Forskolin (parent) was first prepared in DMSO to achieve maximum solubility, followed by dissolving in PBS with a final DMSO concentration known to be safe and non-toxic (1:200). The concentration of Forskolin was 83.33 μM. Noopept was dissolved in the same mixture of PBS containing iso-Forskolin at a final concentration of 146 nM. It is worth noting that the expected final concentrations of Forskolin and Noopept in the CSF following intranasal administration of animals were 10 to 12.5 μM and 20 nM to 25 nM, respectively, with one dose. Three doses of CNS/CT-001 were delivered intranasally over a period of ten days.

The following groups were assigned based on their pharmacological treatments with compounds: (1) WT rats intranasally treated with vehicle (PBS 1X, 1:200 DMSO), PINK1-KO rats intranasally treated with vehicle (1:200 DMSO, at pH 7.5), Forskolin (Fsk), or with CNS/CT-001.

Rabbits: Following the acclimation period, rabbits were segregated into various groups based on their treatments. Each group of rabbits consisted of at least three animals. The rabbits were temporarily restrained in boxes tailored for 10–12 week old rabbits to facilitate the intranasal administration of the compounds. Briefly, the animals were intranasally treated with 30–50 µL of a solution containing Forskolin (100 µM final concentration in CFS) and with Noopept (20 nM) for one dose per nostril by using an applicator or pipettor. Blood was collected at various time points (0.5, 2, 6, and 24 h), while CFS was terminally collected in at least three animals at 4, 12, 24, and 48 h. Following treatments, all animals were analyzed qualitatively and quantitatively for any side effects induced by the compounds administered intranasally (e.g., irritation of the nose, lethality, loss of hair, etc.) and for mobility in their cages. Following the collection of serum at the aforementioned time points, the amount of Forskolin and Noopept was measured from up to 1 mL of processed serum or CSF by performing reversed-phase HPLC as analyzed by performing a standard curve for Forskolin and Noopept in serum spiked with the compounds at increasing concentrations. The pharmacokinetic parameters (half-life, bioavailability, volume of distribution, clearance rate, and others) were determined in the CSF and serum compartments in intranasally-dosed animals using the pharmacokinetic parameters as described below in the “Pharmacokinetic Parameters” section.

*Clarification on the use of rabbits and rats*: While CSF was collected from both rats and rabbits to perform pharmacokinetic analysis of Forskolin and Noopept by HPLC, please note that only IHC analysis of α-synuclein aggregates, TH-positive midbrain dopamine neurons, and Western blot analysis of BDNF and NGF levels was performed in brain tissue derived from rats only.

### 4.4. Preparation of Brain Slices

To perform indirect immunofluorescence assays of brain slices and Western blot assays, the animals underwent anesthesia with isoflurane (5%) followed by transcardial perfusion with 120 mL (10 mL/min) of artificial cerebrospinal fluid solution: 120 mM NaCl, 1.3 mM CaCl_2_, 3.5 mM KCl, 1 mM MgCl_2_, 0.4 mM KH_2_PO_4_, 10 mM Glucose, 5 mM HEPES, at pH 7.4. In order to perform Western blot assays the right hemisphere was saved, whereas the left hemisphere was preserved for performing downstream IHC quantification of midbrain dopamine neurons to analyze for neurodegeneration of midbrain dopamine neurons, and for analysis of α-synuclein aggregates in the cortex, a proxy for detecting the presence of Lewy bodies. In order to locate the prefrontal cortex (PC) and the SN, the specific brain coordinates were used by using a stereotaxic atlas for rats [98] as further described below (Immunofluorescence assay).

### 4.5. Immunofluorescence Assay

To measure the amount of α-synuclein aggregates and the number of dopamine neurons in pharmacologically-treated WT and PINK1-KO rats, the immunostaining of brain slices was performed as previously published [45,46] with some modifications to the protocol. Briefly, following euthanasia of animals by isoflurane administration, the left cerebral hemispheres from intranasally-treated animals were fixed with paraformaldehyde (4% PFA, *w*/*v*) and sequentially treated with increasing concentrations of sucrose solutions (10–30% respectively), followed by embedding the brain tissue in a mixture of Optimal Cutting Temperature (OCT) media with 30% sucrose. The SN or cortical tissues were micro-dissected and then immediately frozen in liquid nitrogen, stored at −80 °C, and sliced using a temperature-controlled cryostat (Leica CM1510-S). The SN was then sectioned into 12 µm brain slices. The stereotaxic coordinates used to locate the SN tissue were: 2.88 mm to 4.32 mm from the interaural line and −6.12 mm to −4.64 mm from Bregma [98]). The SN slices were then placed on pre-coated slides (Superfrost Plus Gold Microscope Slides, Fisher Scientific, Pittsburgh, PA, USA) and stored at −20 °C until ready for downstream processing by IHC. In addition, the first and second layers of the motor cortex (regions M3 and M3) were sectioned as 12 µm brain slices. The cortical slices were then immunolabeled for anti-total α-synuclein or tyrosine hydroxylase (TH) as previously published, with some minor modifications as indicated below.

In brief, midbrain slices were blocked and permeabilized for 1 h by incubating in a solution consisting of 3% bovine serum albumin (BSA) with Triton X-100 in PBS at 0.05% (1X PBS-T), followed by incubation for 2 h with rabbit anti-human TH antibody (Thermo Fisher Scientific, Waltham, MA, USA, P21962, 1:400) in antibody solution (1X PBS, 1.5% BSA). In addition, coronal sections of the prefrontal cortex (PC) mounted as three brain slices/slide were incubated with anti-mouse α-synuclein (BD Transduction Laboratories, San Diego, CA, USA, 6,107,871.5 ug/mL, 2 h at RT). The brain slices were washed at least three times in 1X PBS-T for 10 min. for each wash and incubated with the secondary antibody (Alexa Fluor 647, goat anti-rabbit IgG (1:1000) for midbrain slices; 1:1000 Alexa conjugated anti-mouse IgG -488 for prefrontal cortex slices). To normalize the mean number of midbrain dopamine neurons per epifluorescence micrograph, the nuclei were stained by exposing the tissue with 4′,6-diamidino-2-phenylindole dihydrochloride (DAPI) in a solution containing 70% glycerol at 1.25 µg/mL. The slides were then coverslipped and imaged by using an EVOS-FL Cell Imaging System (Life Technologies, Carlsbad, CA, USA) containing the following filters and setup: GFP (excitation/emission of 470/510 nm), RFP (excitation/emission of 531/593 nm), and Cy5 (excitation/emission of 628/692 nm), at magnifications of 20× (numeric aperture 0.45) or 40× (numeric aperture 0.60) by using a 4X objective.

To measure the degeneration of SN dopamine neurons due to a loss of endogenous PINK1 or to determine the neuroprotective ability of Forskolin or CNS/CT-001 in reversing neurodegeneration of midbrain dopamine neurons in PINK1-KO rats, the mean integrated fluorescence intensity within each region of interest (ROI) was analyzed in TH-stained areas, and analyzed by employing NIH ImageJ software (Bethesda, MD, USA, version 1.50i) as previously published [45,46].

Next, to assess the level of α-synuclein aggregation in the prefrontal cortex, the number of highly-fluorescent puncta or fibrillar structures per epifluorescence field was analyzed by capturing fluorescent images at 10× by using an EVOS-FL epifluorescence microscope (Life Technologies, Carlsbad, CA, USA). It is worth noting that the number of α-synuclein aggregates per epifluorescence field was counted subjectively by an independent observer blind to the experimental conditions. The following exclusion criteria were applied in the analysis in a consistent manner: the α-synuclein puncta were excluded from fibrillar structures, the fluorescent signal was excluded if it localized close to a capillary or a blood vessel or towards the edge of the tissue or near tissue folds, scars, or tears. Finally, the mean number of α-synuclein puncta per epifluorescence field was normalized to the number of DAPI-stained nuclei for the TH immunostaining assays described above.

While CSF was collected from rats and rabbits to perform pharmacokinetic analysis of Forskolin and Noopept by HPLC, it is worth noting that only IHC analysis of α-synuclein aggregates and stereology of TH-positive midbrain dopamine neurons was performed in brain tissue derived from rats only.

### 4.6. Western Blots

Rat brain tissue was homogenized and processed as previously published [43,44] prior to performing Western blot analysis of the levels of BDNF, NGF, and other markers but with the following minor modifications. Following homogenization and preparation of protein lysates, up to 20 µg of protein was electrophoresed on 12% SDS-PAGE gels using the Biorad mini gel system [45,46]. Following the transfer of proteins onto PDVF membranes, the membranes were subsequently blocked in 5% milk with 1X TBS-T (4.6 mM Tris Base, 15 mM Tris HCl, 6.6 M NaCl, 2% Tween-20, pH 7.6), and washed multiple times in 1X TBS-T (5 min/wash). The PDVF membranes were then incubated for 12 h with the primary antibody: anti-human rabbit BDNF (Abcam, Cambridge, UK, 108318; 1:1000), anti-rat goat NGF (AF-556-NA R&D Systems, Minneapolis, MN, USA), and β-tubulin (ab131205, 1:5000, 50 kDa). Following overnight incubation with respective primary antibodies, the PVDF membranes were then incubated with the appropriate horse radish peroxidase (HRP)-conjugated secondary antibodies for at least 2 h at room temperature (25 °C). Finally, the development of immunoreactive bands was then visualized by chemiluminescence detection and analyzed for the levels of immunoreactive bands using Image J software as previously published [7,45,46].

### 4.7. PKA Activity Assay

In order to corroborate the extent that intranasal administration of Forskolin or CNS/CT-001 can elicit an increase in PKA activity in the midbrain of wild-type rats, the PKA activity assay was performed as previously published but with some modest modifications [46]. In brief, rats that were intranasally treated with increasing concentrations of Forskolin for 24 h were anesthetized with isoflurane (5%) and perfused with saline solution. The whole brain tissue was microdissected, harvested, collected, and homogenized to isolate the supernatants as described above, and the amount of protein in the resulting supernatants was quantified by using the BCA kit. The processed samples were stored at −80 °C until used for assessing for PKA activity as described below.

The PKA assay was then performed per the manufacturer’s instructions (Enzo Life Sciences, Farmingdale, NY, USA) as previously published by our research group [41,45,46]. To determine the PKA-specific activity per sample, one of the technical replicates for each brain homogenate per animal was treated with N-(2-{{(2E)-3-(4-Bromophenyl)prop-2-en-1-yl}amino}ethyl)isoquinoline-5-sulfonamide (H-89, 10 µM), a pharmacological inhibitor of PKA. PKA activity was then assessed using the PKA activity assay kit, and the PKA-specific activity was subtracted from the total activity for each sample as published [45,46].

### 4.8. Pharmacokinetic Analyses of Forskolin and Noopept

*Sample processing to analyze Forskolin*: As proof of principle that intranasally-administered Forskolin can cross the BBB, pharmacokinetic analysis of Forskolin in the serum and cerebral spinal fluid (CSF) compartments was performed as further described below. Briefly, 20 μL of Forskolin 166.33 μM was administered intranasally in each rat (3 rats, each weighing 250–300 g, 10 µL per nostril per wild-type rat). Following intranasal administration, the rats were anesthetized, and the CSF and serum were terminally collected. Approximately 1 mL of blood was collected at each time point (0, 0.5, 1, 4, 8, 12, and 24 h) and centrifuged at 2500 rpm for 10 min to separate plasma. The plasma was subsequently collected and transferred to a new tube; then, 200 uL of perchloric acid (PCA) was added for each ml of plasma to precipitate the protein, vortexed for 5 min, and centrifuged for 10 min at 4000 rpm as previously published [99]. The supernatant containing the Forskolin was stored at −80 °C until ready to perform HPLC analysis.

In addition, the CSF was collected at 1, 2, 24, and 48 h following intranasal administration of Forskolin in rats (modified from Nirogi et al., 2016) [100], then 50 µL of PCA was added for 50 µL of CSF, mixed, vortexed 5 min and centrifuged 10 min at 12,000 rpm. The supernatant containing the Forskolin was stored at −80 °C until ready for use. To determine the concentration of Forskolin in plasma and CSF, 20 µL of each sample were chromatographed on a Shimadzu Prominence HPLC fitted with an HC-C18(2), 170 Å, 4.6 × 150 mm reverse phase column (Agilent Technologies, Santa Clara, CA, USA) and an Agilent HC-C18(2) 4.6 × 12.5 mm guard column. Samples were separated using an isocratic mobile phase composed of acetonitrile, methanol, and water (5:1:4 *vol*/*vol*), eluted at 1 mL/min at a temperature of 40 °C as described by Godugu et al., 2016 [101]. In addition, an SPD-M20A photodiode array detector was used to monitor absorbance at 220 nm. The chromatograms were recorded and compared.

*Sample processing to analyze Noopept*: To measure the concentration of Noopept in the plasma and CSF in rabbits, up to 20 µL of each sample were chromatographed on a Shimadzu Prominence HPLC fitted with an Agilent HC-C18(2) at 5 µm, and 10 mm, mobile phase ACN 800:300, 4.6 × 12.5 mm guard column and a wavelength UV detector (LC-290) set to 258 nm at room temperature as previously published [56]. Phosphate buffer was used as the principle buffer (0.02 M, pH 2.7) for performing HPLC with an elution time for Noopept detected at 9.5–10.5 min at a flow rate of 1 mL/min. as previously published [56]. The detection limit for Noopept was measured to be 3.125 µg/mL.

*Pharmacokinetic quantification of Forskolin and Noopept*: The content of both Forskolin and Noopept in chromatographic fractions was determined by using a set of standards to calibrate with respect to the area under peaks for each compound. The pharmacokinetic parameters (half-life, bioavailability, volume of distribution, and others) were determined in the CSF and serum compartments in intranasally-dosed animals using the pharmacokinetic parameters described below. The maximum plasma concentration (C_max_p), maximum CSF concentration (C_max_CSF), time to maximum plasma concentration (T_max_p), and time to maximum CSF concentration (T_max_CSF) was obtained based on experimental observations of standard pharmacokinetic profiles and calculated by using Excel and PRISM as shown in the Appendix A.

### 4.9. Statistical Analyses

Unless otherwise indicated, all behavioral and biochemical data were displayed as mean ± S.E.M. that was compiled from 6–25 animals for in vivo data, or from 3–5 technical replicates for in vitro assays per condition. All behavioral data collected from rats treated with compounds were analyzed using a Student’s *t*-test (two-tailed) for pairwise comparisons. For performing Multiple group comparisons, One-way ANOVA was applied, followed by a Kruskal-Wallis test with a Bonferroni correction test for non-parametric data (in vivo data) by employing GraphPad Prism (version 6.0). Note that all *p*-values less than 0.05 were considered statistically significant.

Sex-related differences: Data from both sexes were displayed for all behavioral studies for the following reason. Given the low number of animals used for each pharmacological condition (5–8 animals for the Forskolin or CNS/CT-001 groups), statistical analyses were performed for both sexes in order to increase the rigor and power for each condition. However, sex-related differences observed between wild-type vs. PINK1-KO rats for motor coordination and grip strength have been published, which showed male PINK1-KO rats exhibited worst motor symptoms and loss of hind limb strength relative to female rats [44].

## 5. Conclusions

In summary, our preclinical data show that intranasal administration of CNS/CT-001 can reverse motor symptoms of PD, the loss of hind limb strength, reduce the number of total levels of α-synuclein aggregates in the cortex and significantly reverse neurodegeneration of midbrain dopamine neurons in symptomatic PINK1-KO rats. Mechanistically, the therapeutic benefits exerted via intranasal administration CNS/CT-001 are associated with significantly increased PKA activity, high brain bioavailability of the active ingredients (Forskolin and Noopept), and increased levels of the neurotrophic factors, including NGF and BDNF in the brain. Overall, our preclinical evidence shows that intranasally delivery of CNS/CT-00, which contains two nootropic agents as active ingredients, is a non-invasive disease-modifying therapy for treating PD for eliciting neuroprotective signaling and neurotrophic support in the brain.

## 6. Patents

The patent (“Disease Modifying Methods for Treating Neurodegenerative Diseases Using Nootropic Agents” authors R.K.D. and R.Y.D.) that protects the intellectual property involving the development, preparation, and method of delivery of CNS/CT-001 to treat PD in patients has been filed by the University of Nevada Reno and pending to be granted. The patent file (US20210220322A1) was published on 22 July 2021.

## Figures and Tables

**Figure 1 ijms-24-00690-f001:**
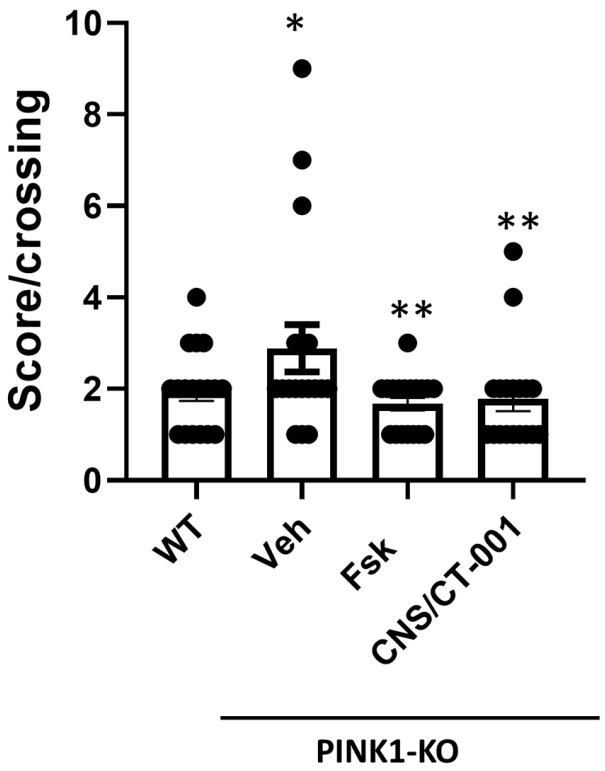
**Intranasal delivery of Forskolin or CNS/CT-001 reverses motor symptoms PINK1-KO rats.** Unbiased assessment of motor coordination as determined by using a beam balance in wild-type (WT) and PINK1-KO rats that were left untreated (baseline) or were intranasally treated with vehicle (Veh) control, Forskolin (Fsk), or with CNS/CT-001 (10 days for five doses). In brief, the data shows that either intranasal administration of Forskolin or CNS/CT-001 efficiently abrogated the loss of motor coordination in symptomatic PINK1-KO rats to a similar extent as in untreated or WT rats. Mean ± SEM. *n* = 12–25 animals per group, *: *p* ≤ 0.05, Kruskal-Wallis test with Bonferroni correction. *: *p* < 0.05 vs. WT group; **: *p* < 0.05 vs. PINK1-KO/Veh group. Each circle indicates an individual animal plotted in the graph.

**Figure 2 ijms-24-00690-f002:**
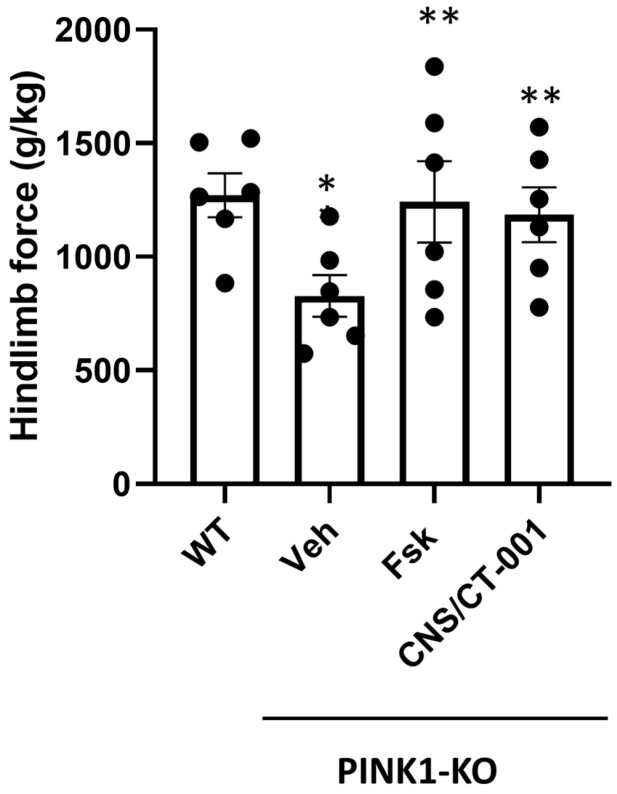
**Intranasal delivery of Forskolin or CNS/CT-001 reverses muscle fatigue in the hindlimbs of PINK1-KO rats.** Compiled quantification of grip strength in the hind limbs in WT and PINK1 KO rats that were left untreated (basal) or intranasally treated with vehicle (Veh) control, Forskolin (Fsk), or with CNS/CT-001 (10 days for five doses). The data are expressed as grams of force per kg. The data show that either intranasal administration of Forskolin or CNS/CT-001 significantly reversed the reduction in hind limb strength in PINK1-KO rats and significantly augmented muscle strength to a similar extent as in untreated or vehicle-treated WT rats. Mean ± SEM. *: *p* ≤ 0.05, Kruskal-Wallis test with Bonferroni correction, *n* = 8–11 animals per group, *: *p* < 0.05 vs. WT group; **: *p* < 0.05 vs. PINK1-KO/Veh group. Each circle indicates an individual animal plotted in the graph.

**Figure 3 ijms-24-00690-f003:**
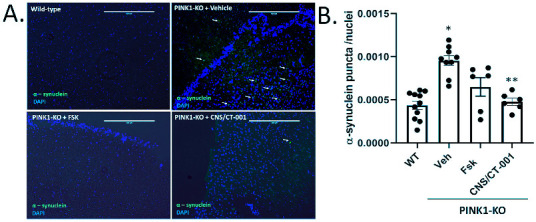
**Intranasal delivery of CNS/CT-001 reverses the amount of α-synuclein aggregates in the cortex of PINK1-KO rats.** (**A**) Representative epifluorescence micrographs of brain slices derived from the M3 and M4 region of the cortex from WT or PINK1-KO rats that were intranasally treated with vehicle control (Veh), with Forskolin (Fsk) or with CNS/CT-001 for ten days, immunostained for total α-synuclein (green) and co-stained with DAPI to label nuclei (blue). The representative arrows point to large α-synuclein puncta (green) within neurons. (**B**) Compiled analysis of the mean number of α-synuclein puncta per epifluorescence field in cortical brain slices derived from WT or PINK1-KO rats that were intranasally treated with vehicle (Veh) control, with Forskolin solution (Fsk) or with CNS/CT-001 for ten days. The data are expressed as the mean number of α-synuclein puncta and normalized to the number of DAPI-stained nuclei. Briefly, the data show that intranasal administration of CNS/CT-001, but not with Forskolin, efficiently reversed the increase in the aggregated α-synuclein in the cortex of PINK1-KO rats to a similar extent as WT rats. Mean ± SEM. *: *p* ≤ 0.05, Kruskal-Wallis test with Bonferroni correction, *n* = 8–11 animals per group, *: *p* < 0.05 vs. WT group; **: *p* < 0.05 vs. PINK1-KO/Veh.

**Figure 4 ijms-24-00690-f004:**
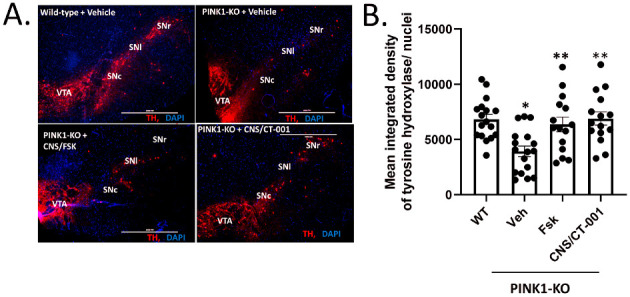
**Intranasal delivery of Forskolin or with CNS/CT-001 reverses the loss of SN neurons in symptomatic PINK1-KO rats.** (**A**) Representative epifluorescence images of SN slices that were immunolabeled for tyrosine hydroxylase (TH; red channel) to visualize SN dopamine neurons from (a) a 10-month-old WT rat that was intranasally treated with vehicle solution or (b) from 10-month-old PINK1-KO rats that were intranasally treated with vehicle solution, or (c) intranasally treated with Forskolin, or (d) intranasally treated with CNS/CT-001. (**B**) Compiled quantification of the mean fluorescence intensity of TH (red channel) in midbrain slices for the indicated treatment conditions (Veh, FSK, CNS/CT-001). Please note that the mean integrated density for the TH-immunostained ROIs within the SN was quantified using Image J and normalized to the number of nuclei labeled with DAPI (blue channel). Mean ± SEM. *: *p* ≤ 0.05, Kruskal-Wallis test with Bonferroni correction, *n* = 14–18 rats per group, *: *p* < 0.05 vs. WT; **: *p* < 0.05 vs. PINK1-KO/Veh. Legends: SNC: substantia nigra pars compacta, SNR: substantia nigra pars reticulata, SNL: substantia nigra pars lateralis. Scale bar: 100 µm. DAPI Blue; TH Red. Each circle indicates an individual animal plotted in the graph.

**Figure 5 ijms-24-00690-f005:**
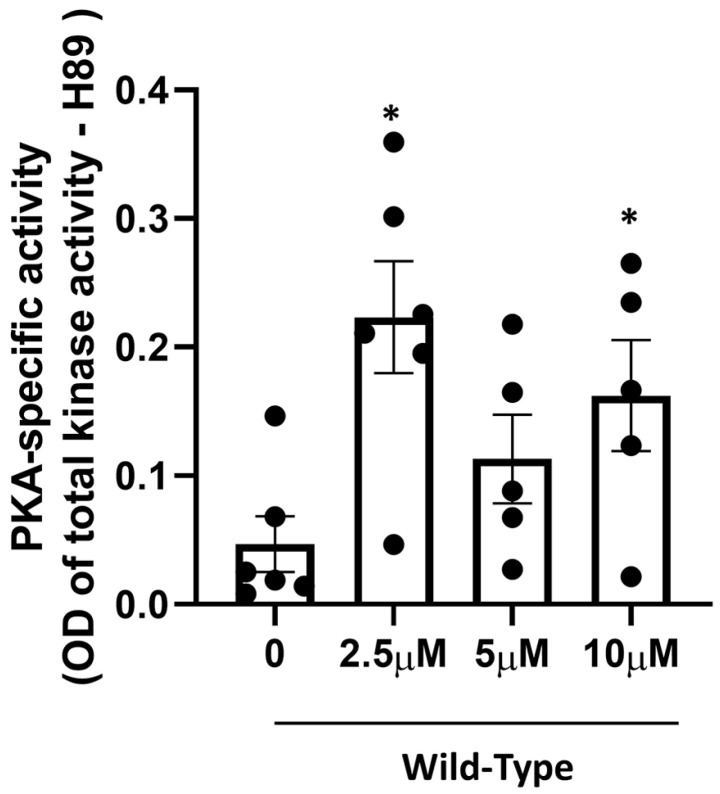
**Intranasal delivery of Forskolin increases PKA activity in the cortex of wild-type rats.** Bar graph showing the mean total PKA activity assessed in the cortex derived from 10-month-old WT treated with vehicle (Veh) or with increasing concentrations of a single dose of Forskolin (2.5, 5, and 10 µM) for 24 h and analyzed for PKA activity by using an ELISA-based kit 24 h following treatments. The data were expressed as PKA-specific activity (total activity minus the activity obtained in samples treated with H89) garnered from at least five animals per condition. Mean ± SEM. *: *p* ≤ 0.05, One-Way ANOVA followed by the Kruskal-Wallis test, 3–6 animals per group). *: *p* < 0.05 vs. 0 µM.

## Data Availability

All of the data (main and supporting data) for this project is available and accessible as displayed in this manuscript and accessible as supporting data (Appendix A) without restrictions.

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
