# Peer review of "Intranasal Administration of Forskolin and Noopept Reverses Parkinsonian Pathology in PINK1 Knockout Rats"

_ijms, 2022, doi:10.3390/ijms24010690_

Round 1

Author Response

We thank the reviewer for taking the time and effort to review our primary research manuscript. We have taken the observations and critique seriously and have addressed all of them in our view as listed below:

1- First of all the manuscript should be checked by an English native speaker to remove the syntax and typos. 

Authors' response: We apologize for any typos and grammatical errors present in the last version of the research manuscript. We have reviewed the manuscript extensively and have revised the typographical errors as marked by using Track Changes in Word.

2- The abstract should be modified to give more digital results rather than elastic sentences. 

Authors' response: We thank the reviewer for this observation. Given that the abstract has a word limit of 200 words, we are only able to modestly expand and edit the abstract to further clarify the extent that intranasal administration CNS/CT001 can significantly reverse symptoms, fatigue and loss of midbrain dopamine neurons in a similar manner as vehicle-treated wild-type rats. While we are unable to provide numerical results in the abstract due to the word limit, the new edits provide the readers a good idea of how the results compared to control-treated animals and an empirical understanding that CNS/CT-001 "completely" reverses the symptoms and PD neuropathology.

3- The introduction should provide more details about the previous work with intranasal delivery of the same molecules. 

Authors' response: We thank the reviewer for this observation. In response to this concern, we have provided more background on research and work related to other nootropic agents in experimental models and in humans. However, it is worth noting that no prior research has ever been conducted regarding the development of an intranasal formulation containing the nootropic agents as described in the present study. This is the first study that characterized the neuroprotective potential of more than one nootropic in combination and that CNS/CT-001 is a first intranasal formulation of its kind that was developed to increase and perpetuate neuroprotective PKA and BDNF signaling the brain.  While similar work on intranasal delivery of the same molecules has not been conducted before, we provided more content in the Introduction section describing the use the oral and intranasal delivery of individual nootropics (Modafinil, and Piracetam) for treating cognitive decline in brain-degenerative disorders in humans (e.g. Alzheimer’s disease). These edits are indicated on page 4 of the Introduction of the revised manuscript.

4- Some sentences are not clear and require rephrasing e.g. Intranasal administration of PINK1-KO 23 rats with CNS/CT-001 ameliorated the loss of midbrain dopamine neurons.

Authors' response: We thank the reviewer for these observation. In response to this concern, we have corrected several sentences that were worded inappropriately and rephrased them in the revised version of the manuscript. In other instances, we opted to break apart some run on sentences into more short declarative sentences to increase flow and readability of the manuscript. These changes are indicated by Track Changes in Word.

5- The novelty of the work should be clearly stated, the aim of the work should be clearly stated; the authors had already investigated the effect of the Forskolin on PD, does the aim is to only investigate the intranasal route? 

Authors' response: We thank the reviewer for this observation and apologize for not making this clear in the previous version of the manuscript. In response to these concerns, we have clearly stated the objective of the study in the Introduction, stated a hypothesis and extensive rationale for developing and using Noopept in the formulation (CNS/CT-001) as indicated by Track Changes in Word.  The present study investigated whether intranasal administration of Forskolin, or Forskolin and Noopept (CNS/CT-001) can reverse motor symptoms and neurodegeneration of midbrain dopamine neurons in Parkinsonian rats. We present additional conceptual novelties of this study. While it is factually correct that intraperitoneal administration of Forskolin has been shown to be sufficient to ameliorate motor symptoms of Parkinson's disease and neurodegeneration of midbrain dopamine neurons in Parkinsonian rats (PINK1-KO rats), Noopept was added in conjunction with Forskolin in the intranasal formulation to facilitate the clearance of alpha synuclein aggregates in symptomatic PINK1-KO rats as a pharmacological approach to help address  cognitive decline in Parkinson's patients in the future. We provided extensive  rationale and justification including Noopept in the same formulation (CNS/CT-001). Specifically, Noopept has been previously shown to be able to reduce the amount of protein aggregation in in vivo experimental models of Alzheimer's disease as well as reducing the amount of alpha synuclein in cell culture models of Parkinson’s disease and in other in vitro studies.  Our study is the first to report an ability of Forskolin and Noopept in reducing alpha synuclein aggregation an in vivo model of Parkinson's disease.  Specifically, our immunohistochemical data show that intranasal administration of CNS/CT-001, but not Forskolin alone, significantly reduces the level of alpha synuclein in the brain of symptomatic PINK1-KO rats in a similar manner as vehicle-treated control (Figure 3). Another novelty is that our study show that the intranasal administration of Forskolin and Noopept (CNS/CT-001) can enhance the level of neurotrophic support in the brain (BDNF and NGF) in a Parkinson's disease model (Supplemental Figure 4). Finally, another innovation is that pharmacokinetic studies show that Forskolin can achieve high bioavailability in the cerebral spinal fluid (>30%) when administered intranasally which is an important finding since Forskolin has low ability to permeate the blood brain barrier if administered orally or intraperitoneally. The new edits which clarify the objective, hypotheses and rationale for this study are located on page 3 and 4 of the revised manuscript (Introduction).

  1. The title of the manuscript stated that the study was conducted using rats, however both rats and rabbits are included which is confusing, in addition the authors should clarify the reason for using rats and rabbits in the introduction or the methodology.

Authors' response: We thank the reviewer for this observation and we apologize for the lack of clarity. In response to this concern, we have edited the methodology of our revised manuscript to further clarify the use of rabbits in the manuscript. Specifically, while the majority of study’s were done in rats (behavioral assays, Western blot and immunohistochemical), we indicated that rabbits were only used to conduct pharmacokinetic studies of Noopept. The rationale for using rabbits is that unlike rats, rabbits are larger and can provide a larger amount of serum and cerebral spinal fluid to conduct pharmacokinetics of Forskolin and of Noopept in the same cohort of animals. In addition, rabbits are more amenable for collecting more cerebral spinal fluid in a humane manner to allow for the detection of two compounds in the same sample which was not possible in rats. Hence, we presented pharmacokinetic data for both Forskolin in rats and rabbits while only pharmacokinetic data of Noopept was obtained in rabbits given the aforementioned technical limitations and animal welfare related concerns of using rats for this type of study. These edits are found on pages 16  of the revised version of the manuscript (Methods section). 

7- Did the authors perform the indirect immunofluorescence assays of brain for rats and rabbits? It is not clear in the methodology 

Authors' response: We thank the reviewer for this observation. To clarify the reviewer's concern, we want to clarify that immunohistochemical studies of brain tissue were only conducted in rats to analyze the extent that CNS/CT-001 can reverse the loss of midbrain dopamine neurons in symptomatic PINK1-KO rats. As eluded in our response to concern #6, rabbits were only used to perform pharmacokinetic analysis of Noopept. We have made the appropriate changes in the Methods section of the revised manuscript. These edits are located on page 18. 

  1. The authors should state the composition of the administrated CNS/CT-001. 

Authors' response: We have indicated the composition of CNS/CT-001 in the Methods section including the working stock concentrations and how Forskolin and Noopept were diluted in the formulation. These edits are found on page 17 of the revised manuscript.

9- The authors should state the full name for any abbreviation for the first time it appears on the manuscript.

Authors' response: We have made the appropriate changes and noted a few instances in which "CNS/CT-001" and other abbreviations (e.g. BDNF and NGF) were not defined the first time in the previous version of the paper. This edit is found on page 3 of the revised manuscript.

10- Too many old references that should be audited.

Authors' response: We thank the reviewer for this observation. However, we want to indicate that the majority of references cited in the paper are recent and not older than 1990  (>90% of references). The references that are older than 1990 are historical studies that were needed to be cited in our study to denote when "Levodopa" was first characterized in humans including its pharmacokinetics, side effects and its pharmacodynamics when administered in oral form. We also cited multiple historical studies that were necessary that characterized the pharmacokinetics and neuroprotective properties of Noopept. However, we have substituted several review papers in the Introduction and Discussion sections of the previous version of the paper with more recent ones as indicated by Track Changes in Word. The more recent reviews that were cited describe the economic burden, and incidence of Parkinson's disease at a global scale and cited more recent reviews regarding the genetics of PD. We hope that the inclusion of more than 7 more recent papers is sufficient to satisfy the editorial requirements of a rigorous yet up-to-date references.

Reviewer 2 Report

The research article titled "Intranasal administration of forskolin and noopet reverses parkinsonian pathology" has scientific value for publication in Int. j of Molecular Sciences, MDPI. But following points need to be clarified by authors.

1. How the abbreviation of the formulation "CNS/CT-001" named, whereas the drugs name are Forskolin and Noopet. What is the meaning of '001'. The meaning of CNS is "Central nervous system". 

2. Line 168: The authors pointed that the intranasal administration of Forskalin is sufficient to reverse motor symptoms and hind limb fatigue in PINK1-KO rats. My doubt is 'why author evaluated the actions of CNS/CT-001. 

3. What is the toxicity profile of CNS/CT-001?

4. How the CNS/CT-001 formulated?

Author Response

  1. How the abbreviation of the formulation "CNS/CT-001" named, whereas the drugs name are Forskolin and Noopept. What is the meaning of '001'. The meaning of CNS is "Central nervous system". 

Authors' response: We thank the reviewer for this observation. We have spelled out and defined the meaning of "CNS/CT-001" in the Introduction of the revised manuscript (page 3 of Introduction section). Specifically, CNS/CT-001 stands for Central Nervous System/ Curative Technologies formulation 001. Additionally, "001" denotes the parent formulation for which other derivatives can be developed in the future.

2. Line 168: The authors pointed that the intranasal administration of Forskolin is sufficient to reverse motor symptoms and hind limb fatigue in PINK1-KO rats. My doubt is 'why author evaluated the actions of CNS/CT-001. 

Authors' response: We thank the reviewer for this concern. we have clearly stated the objective of the study in the Introduction, stated a hypothesis and rationale for developing CNS/CT-001 as indicated by Track Changes in Word (Introduction section, page 4) . While it is factually correct that intraperitoneal administration of Forskolin has been shown to be sufficient to ameliorate motor symptoms of Parkinson's disease and neurodegeneration of midbrain dopamine neurons, we wanted to determine whether both Forskolin and Noopept can reduce the amount of alpha synuclein aggregation in the brain of Parkinsonian rats (PINK1-KO rats).  Therefore, in our study, Noopept was added in conjunction with Forskolin in the intranasal formulation as a nootropic agent to facilitate the clearance of alpha synuclein aggregates in symptomatic PINK1-KO rats as a pharmacological approach to help address  cognitive decline in Parkinson's patients in the future. We provided extensive  rationale of including Noopept in the same formulation (CNS/CT-001). Specifically, Noopept has been previously shown to be able to reduce the amount of beta amyloid  in in vivo experimental models of Alzheimer's disease as well as reducing the amount of alpha synuclein in cell culture models of Parkinson’s disease and in in vitro studies.  Indeed, our immunohistochemical data show for the first time that intranasal administration of CNS/CT-001, but not Forskolin alone, significantly reduces the level of alpha synuclein in the brain of symptomatic PINK1-KO rats in a similar manner as vehicle-treated wild-type rats. Our study is the first to report an ability of Forskolin and Noopept (CNS/CT-001) in reducing alpha synuclein aggregation an in vivo model of Parkinson's disease.  Another rationale for including Noopept with Forskolin in the same intranasal formulation is to enhance and perpetuate neuroprotective PKA signaling and neurotrophic support (BDNF and NGF) in the brain compared to Forskolin alone. The rationale for doing so is that both Forskolin and Noopept can directly/indirectly elevate PKA signaling and BDNF by targeting adenylate cyclase (Forskolin) and AMPA receptors (Noopept). Hence,  our study show that the intranasal administration of Forskolin and Noopept (CNS/CT-001) can enhance the level of neurotrophic support in the brain (BDNF and NGF) in a Parkinson's disease model to a greater extent than Forskolin alone, although in a non-significant manner compared to Forskolin alone (Supplemental Figure 4).  Overall, our data suggest that intranasal administration of Forskolin ameliorates symptoms of Parkinson's disease, fatigue and neurodegeneration of dopamine neurons whereas Noopept facilitates clearance of alpha synuclein aggregates and elevates neurotrophic support in Parkinsonian rats. The rationale for including Noopept and testing CNS/CT-001  is included in the edits of the revised version of the manuscript.

3.What is the toxicity profile of CNS/CT-001?

Authors' response: We thank the reviewer for this observation. However, we want to clarify that we have not conducted a rigorous and high resolution study that characterizes the toxicity of CNS/CT-001 in various animal models. However, in the revised version of the manuscript, we have indicated that descriptive and observational studies suggest that intranasal administration of CNS/CT-001 (5 doses for over 10 days) does not induce noticeable irritation or inflammation of the nasal passages in rats or rabbits treated with CNS/CT-001 nor we observed any adverse events including significant loss of weight loss, discoloration of fur or death of the animals. However, it is worth noting that Noopept is commercially available for human consumption as well as low concentrations of Forskolin (.e.g. herbal extract or supplement) and so far, no major adverse events have been documented in humans and only cause minor symptoms (nausea, low blood pressure, diarrhea by Forskolin or high blood pressure and headaches by Noopept). However, side effects are only seen after prolonged consumptions and at higher doses in humans. Overall, future  rigorous toxicological studies are beyond the scope of the present paper given the limited resources and time needed to conduct rigorous toxicological profiling.  We made the appropriate edits on page 6 and 11 of the revised Results section.

4. How the CNS/CT-001 formulated?

Authors' response: We thank the reviewer for this concern. We have indicated the composition of CNS/CT-001 in the Methods section of the revised version of the manuscript including the working stock concentrations and how Forskolin and Noopept were diluted in the formulation. These edits are indicated on page 17 of the Methods section of the revised manuscript.

Round 2

Reviewer 1 Report

The authors had performed all the necessary changes and the manuscript can be accepted for publication  

Congratulations